# Intention to Vaccinate against COVID-19 in Adolescents: A Systematic Review

**DOI:** 10.3390/vaccines11081393

**Published:** 2023-08-21

**Authors:** Shyn Yi Tan, Prawira Oka, Ngiap Chuan Tan

**Affiliations:** 1Lee Kong Chian School of Medicine, Nanyang Technological University Singapore, Singapore 308232, Singapore; tans0390@e.ntu.edu.sg; 2SingHealth Polyclinics, Jalan Bukit Merah Connection One, Singapore 150167, Singapore; tan.ngiap.chuan@singhealth.com.sg; 3SingHealth-Duke NUS Family Medicine Academic Clinical Programme, Outram Road, Singapore 169608, Singapore

**Keywords:** adolescents, barriers, COVID-19, enablers, public health, vaccines

## Abstract

Background: Multiple COVID-19 vaccines have been approved for use in adolescents; these vaccines play a critical role in limiting the transmission and impact of COVID-19. This systematic review aims to summarize the willingness of adolescents aged 10 to 19 years to receive the COVID-19 vaccination and the factors influencing their decision. Methods: A search of literature published between January 2018 and August 2022 was performed in Medline©, EMBASE©. and CINAHL© electronic databases. Studies published in English that assessed adolescents’ intentions to receive the COVID-19 vaccine were included. Qualitative studies and those unrelated to the COVID-19 vaccine were excluded. The study was conducted based on the PRISMA guidelines. Results: Of the 1074 articles retrieved, 13 were included in the final review. Most studies were conducted in the US (*n* = 3) and China (*n* = 3). The pooled prevalence of COVID-19 vaccine acceptance among adolescents was 63% (95% CI: 52–73%). Factors influencing intent to vaccinate were divided into five categories: “Socio-demographic determinants”; “Communication about COVID-19 pandemic and vaccination”; “COVID-19 vaccine and related issues”; “COVID-19 infection and related issues” and “Other determinants”. The enablers were sociodemographic factors including older age, higher education level, good health perception, and parental norms in terms of parental vaccination acceptance; perceived vaccine effectiveness and safety; a desire to protect themselves and others; recent vaccination; and anxiety. The barriers were concerns over vaccine effectiveness, safety, and long-term side effects; low perceived necessity and risk of infection; and needle phobia. Conclusions: This review highlighted that adolescents’ intent to vaccinate is driven by a desire to protect themselves and others. However, concerns over vaccine effectiveness, safety, and long-term side effects hinder COVID-19 vaccine uptake. To improve vaccination acceptance, policymakers should address adolescents’ concerns via more targeted public health messaging, while schools should leverage peer norms to positively influence vaccination intent.

## 1. Introduction

The highly contagious coronavirus 2019 (COVID-19) continues to pose a global health threat, with transmission and mutations contributing to increased morbidity and mortality [1]. As of February 2023, more than 755 million people have been infected with COVID-19 [2], with children and adolescents representing approximately a fifth of these cases [3].

The COVID-19 vaccine has been proven to be safe in adolescents with predominantly mild to moderate adverse reactions such as injection-site pain, headache, fever, and fatigue. Adolescents also mounted a stronger or non-inferior immune response to the vaccine when compared to adults [4,5,6]. Despite this, distrust in the COVID-19 vaccine persists among adolescents [7], with a resultant low vaccine uptake.

Vaccine hesitancy in adolescents is further exacerbated by their perception that they are at lower risk of infection [8,9]. Their belief contrasts with reports that adolescents, particularly those unvaccinated, were more susceptible to COVID-19 than older adults [10,11]. Furthermore, unvaccinated adolescents were also six times more likely to require hospitalization [12] and experience long-term COVID-19-related complications than their vaccinated counterparts [13,14,15,16].

With the COVID-19 vaccine efficacy rate at 71.2% in adolescents [17], vaccine-induced herd immunity requires coverage of 90% or higher [18,19,20]. As of June 2023, according to the CDC, only 59.9% of American adolescents are currently vaccinated with at least one dose of the COVID-19 vaccine, which is grossly inadequate [21]. Vaccination coverage for adolescents in other countries is also severely lacking [22,23].

Studies conducted globally have focused primarily on vaccine uptake among adults [24,25,26,27]. Low vaccine uptake among adolescents has been attributed to peer pressure and sociodemographic characteristics [28,29]. More countries have begun to extend vaccination to adolescents. As parental consent is required for vaccination in adolescents, previous studies have focused on the parents’ intentions to vaccinate their adolescent children [30,31,32,33]. It is important to involve adolescents in the medical decision-making process, with a US study finding that they wish to be responsible for their own medical decisions [34]. Therefore, the adolescent perspective on understanding the enablers and barriers to vaccine acceptance will allow healthcare professionals and policymakers to develop targeted strategies to increase vaccine acceptance to combat COVID-19 and prepare for future pandemics.

### Aim

This systematic review aims to determine the willingness of adolescents to receive the COVID-19 vaccine and to identify enablers and barriers to vaccine acceptance.

## 2. Methods

A protocol detailing the search methods employed was registered on PROSPERO (CRD42022351291).

### 2.1. Data Sources

A comprehensive literature search of articles published between January 2018 and August 2022 was conducted using these electronic databases: PubMed/MEDLINE, EMBASE, and CINAHL. Google Scholar was also screened to avoid missing relevant articles.

### 2.2. Search Strategy

The search strategy combined the following MeSH terms and free text: “COVID-19” [MeSH] OR “Vaccination” [MeSH] OR “Adolescents” [MeSH] OR “Willingness”. The detailed search strategies employed in Pubmed, Medline, EMBASE, and CINAHL can be found in Appendix A.

### 2.3. Data Collection and Study Selection

All studies obtained from the listed databases were exported to EndNote, and duplicates were removed prior to screening. Two researchers (PO and SYT) independently screened the titles and abstracts of the exported studies according to the inclusion and exclusion criteria. The full text of abstracts and titles was extracted for review if they were ambiguous. Full texts were then extracted for all remaining studies using the Nanyang Technological University Library and assessed for eligibility.

### 2.4. Inclusion Criteria

Observational studies published in English involving participants aged 10 to 19 years and their intention to receive the COVID-19 vaccine as defined by vaccine acceptance, willingness, or hesitancy were included.

### 2.5. Exclusion Criteria

Non-observational studies, such as purely qualitative studies and those unrelated to the COVID-19 vaccine, were excluded.

### 2.6. Data Extraction

Two reviewers (PO and SYT) independently conducted the extraction, with discrepancies resolved by a third researcher (NCT). The findings of interest were study design, study population, reported outcomes (vaccine willingness/acceptance, or hesitancy), enablers, and barriers to intention to receive the COVID-19 vaccine. Enablers and barriers refer to factors that positively and negatively contribute to the intention to receive the COVID-19 vaccination, respectively.

### 2.7. Risk of Bias Appraisal

The included studies were independently appraised by the two researchers (PO and SYT), with any discrepancies resolved by a third researcher (NCT). The selected articles were assessed using the Joanne Briggs Institute (JBI) checklist for prevalence studies. This tool was selected as a preliminary literature review reveals a preponderance of cross-sectional studies. The outcome of the assessment can be found in Table 1.

### 2.8. Publication Bias

A search of the gray literature performed through WorldCat^®^ and Google Scholar did not yield any relevant unpublished work. Visualisation of funnel plot and Egger’s regression test was also performed.

### 2.9. Data Synthesis

The full text of all included articles was analyzed by two researchers (PO and SYT). Key findings from the included studies were narratively synthesized. The following study characteristics were included: study location, study timing, study population, study design, survey instruments, reported outcomes, and limitations.

### 2.10. Data Analysis

Study characteristics and the outcome of interest were narratively summarized in tables. To facilitate our meta-analysis, the formula “vaccine hesitancy = 1 − vaccine acceptance” was used. The pooled prevalence of vaccine acceptance and hesitancy was performed using R statistical software version 4.3.1.

## 3. Results

A total of 1763 studies were identified through the search strategies of PubMed/MEDLINE, EMBASE, CINAHL, and Google Scholar. After excluding duplicates, 1074 articles remained. Titles and abstracts were screened, resulting in the exclusion of 1403 articles. The full text was sought and reviewed for 31 articles. There were 13 articles that satisfied the eligibility criteria and were included in this systematic review (Figure 1).

### 3.1. Study Characteristics

All the included studies, published up to 24 August 2022, assessed the intention of adolescents to receive the COVID-19 vaccine and its associated factors, including attitudes, opinions, and perspectives. All the studies were cross-sectional, and adolescents were recruited either from existing databases or through convenience or snowball sampling through online outreach. The studies originated from various parts of the world, including Asia [37,38,39,41,42,46,47], the United States (US) [43,44,45], and Europe [35,36,40]. All studies employed questionnaires as study instruments.

### 3.2. Intention to Vaccinate against COVID-19 in Adolescents

The majority of studies (*n* = 8) reported intention to vaccinate in terms of both vaccine acceptance and hesitancy rates [35,36,37,41,42,43,44,47]. Four studies only reported vaccine acceptance [38,39,40,46], and one study only reported vaccine hesitancy [45].

The vaccine acceptance rates among adolescents ranged from 35.5% in Russia [37] to 94.3% in China [39] (Table 2). Encouragingly, eleven of the studies described vaccination acceptance rates of more than 50% [35,36,37,38,39,40,41,42,43,44,47].

Vaccine hesitancy rates (VHR) ranged from 5.7% in China [39] to 64.5% in Russia [37]. Then the two lowest VHRs originated from China [39,47].

The pooled prevalence of vaccine acceptance is 63% (95% CI: 52–73%); there was significant heterogeneity (I^2^ = 99.9%) (Figure 2). Vaccine acceptance rates were highest in Asia (72%, 95% CI: 54–85%) compared to Europe (55%, 95% CI: 39–70%) and the United States (52%, 95% CI: 40–63%).

### 3.3. Publication Bias

There was no significant funnel plot asymmetry; regression-based Egger’s test for small-study effects did not indicate evidence of publication bias (*p* = 0.545) (Figure 3).

### 3.4. Factors Influencing Intention to Vaccinate against COVID-19 in Adolescents [48]

Factors were classified according to a modified conceptual framework of factors influencing vaccine acceptance and hesitancy proposed by Joshi et al. [48]. The factors were first dichotomized into enablers (Table 3) and barriers (Table 4) before being subclassification into five broad categories. The categories included: “Socio-demographic determinants”; “Communication about COVID-19 pandemic and vaccination”; “COVID-19 vaccine & related issues”; “COVID-19 infection & related issues” and “other determinants”.

Nine studies reported both enablers and barriers influencing the intention to vaccinate [35,36,38,40,42,43,44,46,47]. Of the remaining studies, one study solely recorded enablers [39], two studies recorded barriers alone [41,45], and a final study did not report any factors influencing vaccination intention [37].

### 3.5. Socio-Demographic Determinants

The most commonly identified socio-demographic enablers included parental norms in terms of their own acceptance of COVID-19 vaccination [39,43,44,46], parental wishes for an adolescent to be vaccinated [35,39,43], higher education level [35,38,39], older age [36,39,43], and good subjective health perception [38,47].

Although the female gender was identified as a barrier to vaccine acceptance in Sweden [40] and China [42], multiple other studies [35,38,39,43,45,46] found no association, while Fazel et al. found that being female was instead an enabler to vaccine acceptance [36].

### 3.6. Communication about COVID-19 Pandemic and Vaccine Related Factors

Adolescents cited possessing information about vaccine safety [44] and wanting to follow governmental recommendations [35] as reasons to be vaccinated.

On the other hand, lack of doctor recommendation [42], uncertainty over the vaccination application process [41], lack of access to vaccination-related information [47], and vaccination conspiracy theories [47] were cited as barriers to vaccination.

### 3.7. COVID-19 Vaccine-Related Factors

Multiple studies recognized perceived vaccine effectiveness [35,38,39,42,44] and safety [35,38,39,44] as enablers of COVID-19 vaccine acceptance among adolescents.

Conversely, barriers to vaccine acceptance included concerns over COVID-19 vaccine safety [35,38,41,43,46,47] and effectiveness [35,41,43,46,47], unknown long-term consequences [35,41,46,47], low perceived necessity [35,40,41,43], needle phobia [38,41], and fear of vaccine causing COVID-19 infection [43]. Notably, adolescents with high perceived knowledge of the COVID-19 vaccine were less inclined to take the vaccine [38].

### 3.8. COVID-19 Infection-Related Factors

Adolescents cited wishing to protect others [35,38,40,42,44,46] and themselves [38,42,44,46] as the main drivers behind their intent to vaccinate. Other identified enablers were life being affected by COVID-19 [39,46] and a higher perceived risk [38,42] and severity of COVID-19 infection [38,44].

Barriers to vaccination include a low perceived risk of COVID-19 infection [42,44,47] and severe COVID-19 infection [40,49] and the belief that public measures are sufficient to prevent transmission [38,46].

### 3.9. Other Factors

Adolescents mentioned returning to a pre-COVID-19 lifestyle [35,38,44] and relieving public health measures as reasons for their positive vaccination intention [38,40,44]. Increased anxiety (both unrelated and due to COVID-19) [36,40,43], recent childhood vaccination history [38], and influenza vaccination within the past year [46] were associated with vaccine acceptance.

Being less socially connected (in terms of increasing media usage and less identification with the school community) [36,45] also had a negative influence on their vaccination intention.

### 3.10. Risk of Bias in Studies

The studies were appraised using the JBI checklist for prevalence studies in Table 1. Most studies failed to employ validated methods to assess vaccination intention [35,38,39,41,43,44,46,47] and did not clearly discuss response rates and reasons for non-response [35,36,37,38,40,41,44,46,47].

## 4. Discussion

Overall, the willingness to be vaccinated varies extensively across populations, but adolescents in most countries studied appeared to be receptive to COVID-19 vaccination.

Vaccine acceptance rates (VAR) were above 50% for 11 of the studies [35,36,37,38,39,40,41,42,43,44,47], with only adolescents from Russia, Hongkong, and the United States reporting a VAR of 35.5% [37], 38.6% [46], and 42% [45], respectively. China had the highest VAR of 60.1%, 83.5%, and 94.3% [39,42,47]. The Chinese study that reported a VAR of 60.1% performed their data collection in December 2020 [42]. In contrast, the remaining two Chinese studies performed their data collection in March and August 2021 [39,47]. The time elapsed between studies could account for the improved VAR due to the increased availability of vaccines and vaccine-related information.

Vaccine hesitancy rates (VHR) were lower in Asia, with the lowest rates in China [39,47]. Among Chinese studies, there was a reduction in VHR from 39.9% to as low as 5.7% relative to the period of data collection [39,42,47].

The main enablers of vaccine acceptance were confidence in vaccine effectiveness and safety; the desire to protect others and themselves; and parental acceptance of the COVID-19 vaccine. The result suggests that parental norms strongly influence the vaccination intention of adolescents, which is congruent with existing literature [50,51,52]. It emphasizes the need for strategies to target vaccine hesitancy in both parents and their children. Oka et al. noted that the main sources of information on COVID-19 vaccines for adolescents originated from family members [41]. Therefore, local frontline community healthcare professionals could also raise adolescents’ and parents’ awareness of their role in curbing the virus spread by disseminating evidence-based information on vaccine safety and effectiveness. Such personalized measures can assist in clearing their doubts and addressing these major concerns [53].

The main barriers to vaccine acceptance were concerns over vaccine safety and efficacy, concerns over long-term side effects, and a low perceived necessity. The adolescent concerns are consistent with current literature [7,54,55] and could stem from delayed approval of vaccines for adolescents [56]. Adolescents could also be more skeptical regarding vaccine effectiveness and safety due to the exaggerated information they encounter on social media [57,58,59]. Despite several randomized controlled trials proving the safety and effectiveness of the COVID-19 vaccine [4,5,6,60,61,62], such evidence might not have been conveyed to the public, especially adolescents, in lay terms that this target audience could comprehend. Hence, public health messages on vaccine safety should be packaged to catch the attention of adolescents or their parents and help them understand both the benefits and potential risks for informed decision-making.

The low perceived necessity of the COVID-19 vaccine among adolescents could result in low vaccine acceptance. Studies examining adolescent brain development suggest that their flawed risk perception may be due to their relatively underdeveloped prefrontal cortex as well as their decreased ability to anticipate future consequences [63,64,65,66]. While acknowledging that adolescents may not act in their best interests, it is important to provide them with the appropriate information and resources to make their own decisions.

This review had several limitations. First, only articles published in English were included. Articles from non-English-speaking countries may have been published in their native language; thus, adolescents from those countries may not be represented in this paper.

Secondly, most studies did not use validated instruments to record associated factors influencing vaccine acceptance. Although meta-analysis was performed, the high heterogeneity of the data limited the quality of the analysis. Employing validated instruments that are standardized across studies would facilitate the generation of higher-quality data.

Thirdly, a causal relationship between the intention to vaccinate and the identified factors could not be established due to the cross-sectional design of the included studies. There are likely multiple confounders influencing vaccine acceptance, and a randomized controlled trial could be conducted to definitively identify causative factors. However, it is impractical and impossible to conduct one as there are too many factors to control. Nonetheless, when adequately powered, cross-sectional studies are more practical and provide sufficient data.

Fourthly, the adolescent age group, ranging from 10 to 19 years, was selected based on the WHO definition. In many countries, adolescents aged 10 to 17 years are considered children and require parental consent for vaccination, while adolescents aged 18 to 19 years are considered adults and can act based on their own opinions. This difference in perspective was not accounted for, as the included studies only performed analyses on the whole population with no subgroup analyses. Future studies could target only younger adolescents or perform subgroup analyses in studies with older adolescents.

Finally, each country had differing COVID-19 vaccine approval and procurement timelines. The varied rollout among adolescents potentially affected their views and intentions to vaccinate. Vaccine type and brand were also rarely mentioned as potential confounding factors. These logistical considerations could be further explored in prospective research to determine if they play a significant role in influencing vaccine acceptance.

Nevertheless, this is the first systematic review to examine COVID-19 vaccine acceptance rates and associated reasons for vaccine uptake among adolescents. This review only included papers that reported the adolescent’s perspective and excluded those that examined parental vaccine hesitancy. Both quantitative and qualitative outcomes were reported, facilitating a deeper understanding of the adolescent thought process behind their intention to vaccinate.

Future research should conform to the SAGE working group definition of vaccine hesitancy and utilize validated instruments. Reduced heterogeneity will facilitate the conduct of future meta-analyses to better evaluate the intention to vaccinate and the associated factors.

## 5. Conclusions

This systematic review adds evidence on the prevalence, enablers, and barriers that influence vaccination intention in the adolescent population. The pooled prevalence of COVID-19 vaccine acceptance among adolescents was 63% (95% CI: 52–73%). Concerns about vaccine safety and effectiveness remain a major concern, which should be adequately addressed to expedite vaccine uptake. Generating peer norms and obtaining parental concordance are ways to enhance vaccination intention among adolescents.

## Figures and Tables

**Figure 1 vaccines-11-01393-f001:**
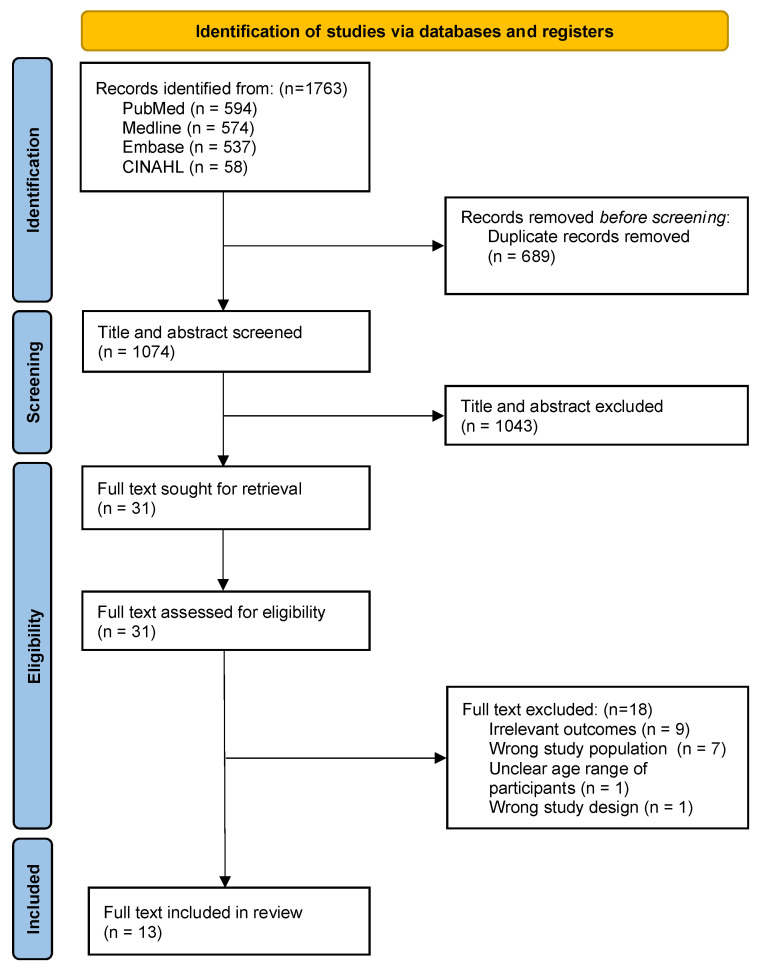
PRISMA flow diagram for included studies.

**Figure 2 vaccines-11-01393-f002:**
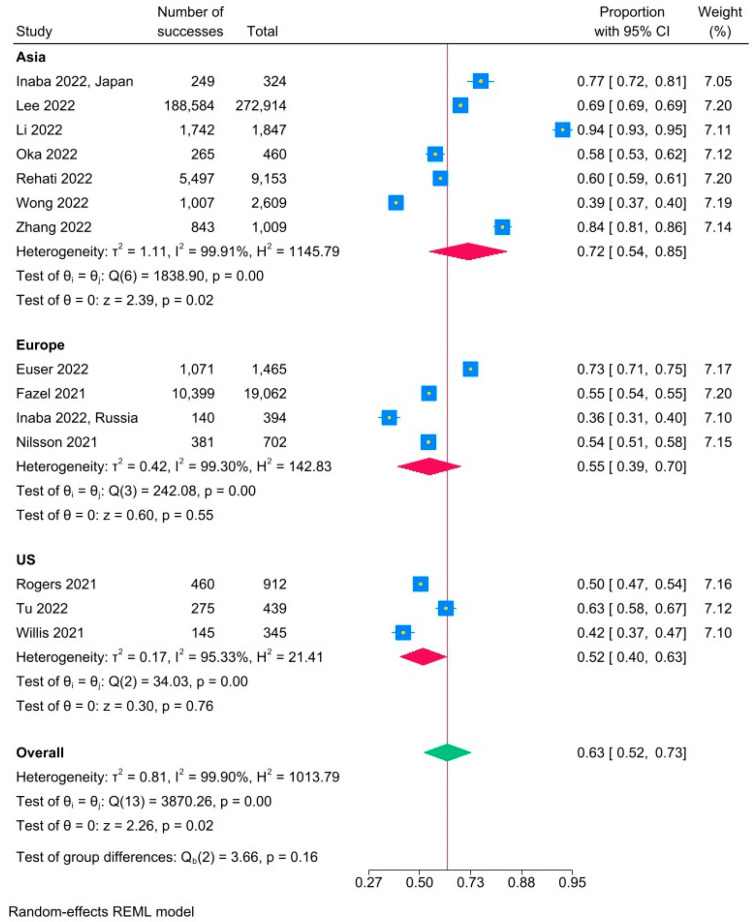
Pooled Prevalence of Vaccine Acceptance by Region [35,36,37,38,39,40,41,42,43,44,45,46,47].

**Figure 3 vaccines-11-01393-f003:**
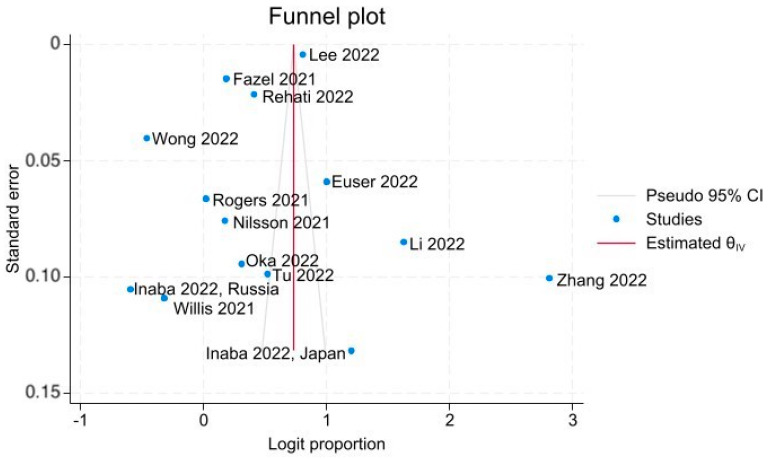
Funnel Plot of COVID-19 Vaccine Acceptance with 95% CI [35,36,37,38,39,40,41,42,43,44,45,46,47].

**Table 1 vaccines-11-01393-t001:** Included studies graded using the Joanna Briggs Institute’s (JBI) Critical Appraisal Checklist for Prevalence Studies.

No	Study	Q1	Q2	Q3	Q4	Q5	Q6	Q7	Q8	Q9	Overall Appraisal
1	Euser et al., 2021 [35]	Y	Y	Y	U	Y	N	Y	Y	N	Include
2	Fazel et al., 2021 [36]	Y	Y	Y	Y	N	Y	Y	Y	N	Include
3	Inaba et al., 2022 [37]	N	Y	Y	Y	N	Y	Y	N	N	Include
4	Lee et al., 2022 [38]	Y	Y	Y	Y	Y	N	Y	Y	N	Include
5	Li et al., 2022 [39]	Y	U	Y	Y	Y	N	Y	Y	Y	Include
6	Nilsson et al., 2021 [40]	Y	N	Y	N	Y	Y	Y	Y	N	Include
7	Oka et al., 2022 [41]	Y	N	Y	Y	N	N	Y	N	N	Include
8	Rehati et al., 2022 [42]	Y	Y	Y	Y	Y	Y	Y	Y	Y	Include
9	Rogers et al., 2021 [43]	Y	Y	Y	Y	Y	N	Y	Y	Y	Include
10	Tu et al., 2022 [44]	Y	Y	Y	Y	Y	N	Y	Y	N	Include
11	Willis et al., 2021 [45]	Y	N	Y	Y	Y	Y	Y	Y	Y	Include
12	Wong et al., 2022 [46]	Y	Y	Y	Y	Y	N	Y	Y	N	Include
13	Zhang et al., 2022 [47]	Y	Y	Y	Y	Y	N	Y	Y	N	Include

Each study will be assessed for quality with the JBI by two independent authors. Differences in grading were resolved by an arbitrator. The final grading of each article on each question is shown here. Y: Yes, N: No, U: Unclear, SFI: Seek Further Info.

**Table 2 vaccines-11-01393-t002:** Summary of Cross-sectional Studies Selected for Analysis.

Author	Period of Data Collection	Population	Age (Years)	Country	Instrument	Outcome	
						Vaccine Acceptance	Vaccine Hesitancy
Euser 2022 [35]	June 2021	1465	16–17	Netherlands	self-designed questionnaire	73.1%	26.9%
Fazel 2021 [36]	May to July 2021	33,556 ^	9–18	England	Oxford COVID-19 vaccine hesitancy scale, Revised Children’s Anxiety and Depression Scales (RCADS), and Bird Checklist of Adolescent Paranoia (B-CAP)	54.6%	45.4%
Nilsson 2021 [40]	July to November 2020	702	15–19	Sweden	Adapted questionnaire and numerical rating scale to assess anxiety	54.3%	45.7%
Inaba 2022 [37]	May 2021	394 (Russia)	15	Russia	self-designed questionnaire	35.5%	64.5%
Inaba 2022 [37]	July 2021	327 (Japan)	15	Japan	self-designed questionnaire	76.9%	23.1%
Lee 2022 [38]	June to July 2021	272,914	12–17	Korea	self-designed questionnaire	69.1%	30.9%
Li 2022 [39]	August to October 2021	1847	12–17	China	self-designed questionnaire	94.3%	5.7%
Rehati 2022 [42]	December 2020	9153	12–17.5	China	self-designed questionnaire based on the health belief model	60.1%	39.9%
Zhang 2022 [47]	March to April 2021	2414 *	16–21	China	self-designed questionnaire with Psychosocial Index-Young (PSI-Y) and Social Support Rating Scale (SSRS)	83.5%	16.5%
Oka 2022 [41]	June to November 2021	460	16–17	Singapore	face validated self-designed questionnaire	57.6%	42.4%
Wong 2022 [46]	June 2021	2609	12–18	Hong Kong	self-designed questionnaire	38.6%	61.4%
Rogers 2021 [43]	June 2021	916	12–17	United States	self-designed questionnaire	50.4%	49.6%
Tu 2022 [44]	October to November 2021	439	13–17	United States	self-designed questionnaire based on protection motivation theory	62.6%	37.4%
Willis 2021 [45]	May 2021	345	12–15	United States	self-designed questionnaire	42%	58%

* only data from older adolescents (*n* = 1009) aged 16 to 17 years old were included. ^ only data from adolescents (*n* = 19,062) aged 12 to 18 years old were included.

**Table 3 vaccines-11-01393-t003:** Enablers of COVID-19 Vaccine Acceptance.

Category	Factor	Number of Studies	References
Socio-demographic	Parental norms (parental acceptance of COVID-19 vaccination)	4	[39,43,44,46]
	Parental wishes for adolescents to be vaccinated	3	[35,39,43]
	Higher education level	3	[35,38,39]
	Older age	3	[36,39,43]
	Subjective health perception	2	[38,47]
	Male gender	1	[40]
	Female gender	1	[36]
	Peer norms	1	[43]
	Being from rural area	1	[39]
	Higher parental education	1	[43]
	Higher household income	1	[43]
	Asian American or Latinx ethnicity	1	[43]
Communication about COVID-19 pandemic and vaccination	Possessing information about vaccine safety	1	[44]
	Want to do what is best according to the government	1	[35]
COVID-19 vaccine and related issues	Confidence in vaccine effectiveness	5	[35,38,39,42,44]
	Confidence in vaccine safety	4	[35,38,39,44]
	Perceived risk-benefit of vaccine	1	[38]
COVID-19 infection and related issues	To protect others	6	[35,38,40,42,44,46]
	To protect themselves	4	[38,42,44,46]
	Life affected by COVID-19	2	[39,46]
	Perceived risk of COVID-19	2	[38,42]
	Perceived severity of COVID-19	2	[38,44]
	Previous quarantine due to COVID-19	1	[39]
	Not living with someone with COVID-19	1	[40]
	Knowing someone with COVID-19	1	[46]
Others	Return to Pre-COVID-19 lifestyle	3	[35,38,44]
	Relieve public health measures	3	[38,40,44]
	Increased Anxiety (including COVID-19 related anxiety)	3	[38,40,44]
	Influenza vaccination in past year	1	[46]
	Recent childhood vaccination history	1	[38]

**Table 4 vaccines-11-01393-t004:** Barriers to COVID-19 Vaccine Acceptance.

Category	Factor	Number of Studies	References
Socio-demographics	Female gender	2	[40,42]
	Being from urban city	1	[42]
	Neither parent born in UK	1	[36]
	Staying in boarding school	1	[42]
	Lower socioeconomic status	1	[36]
	History of physical disease	1	[47]
	Lifestyle (smoking, less exercise)	1	[36]
Communication about COVID-19 pandemic and vaccination	Unsure of vaccination application process	1	[41]
	Lack of doctor recommendation	1	[42]
	Lack of access to vaccine-related information	1	[41]
	Vaccination conspiracy theories	1	[41]
COVID-19 vaccine and related issues	Concerns over vaccine safety	6	[35,38,41,43,46,47]
	Concerns over vaccine effectiveness	5	[35,41,43,46,47]
	Concerns over long term side effects of vaccine	4	[35,41,46,47]
	Low perceived necessity	4	[35,40,41,43]
	Needle phobia	2	[38,41]
	Fear it may cause them to be infected with COVID-19	1	[43]
	High perceived knowledge of COVID-19 vaccine	1	[38]
	Cost concerns	1	[42]
	Belief that natural immunity is better than vaccination	1	[43]
	Unpleasant vaccination experience	1	[41]
COVID-19 infection and related issues	Low perceived risk of infection	3	[42,44,47]
	Low perceived risk of severe COVID-19 infection	2	[40,49]
	Public measures sufficient to prevent COVID-19 infection	2	[38,46]
	Fear it may affect COVID-19 swab test results	1	[41]
	Lack of information about COVID-19	1	[42]
	Belief that COVID-19 did not influence their lives	1	[42]
Others	Less social connection	2	[36,45]
	Not refraining from their normal social activities or group training	1	[40]
	Do not want interruption to school and studies	1	[38]

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
