# Peer review of "Intention to Vaccinate against COVID-19 in Adolescents: A Systematic Review"

_vaccines, 2023, doi:10.3390/vaccines11081393_

Round 1
Reviewer 1 Report
Q1. The period of the search for the eligible studies should be defined. From XXX to August 2022?
Q2. I think that you should better describe the search keywords. How did you search those terms on Pubmed, etc?
Q3. It is not clear to me why did you not perform a metanalysis to estimate a pooled prevalence of hesitancy. There are several statistical methods to deal with high eterogenity. doi: 10.1080/14760584.2022.2100766.
Q4. In conclusions (Comparison with existing literature) you speak about "association", but your review cannot estimate an association. Please revise
Q5. In introduction, you should describe the characteristics of the vaccination in adolescence (efficacy, effectiveness, immunogenicity, safety).
Author Response
Thank you for your helpful comments.
Comments and Suggestions for Authors
Q1. The period of the search for the eligible studies should be defined. From XXX to August 2022?
Thank you for your feedback, this segment has been updated. (Line 74-76)
Q2. I think that you should better describe the search keywords. How did you search those terms on Pubmed, etc?
Thank you for your feedback. As the search strategy is rather extensive it was not included in the main text. The text has been amended to provide better clarity. (Line 78-81)
Q3. It is not clear to me why did you not perform a metanalysis to estimate a pooled prevalence of hesitancy. There are several statistical methods to deal with high heterogeneity. doi: 10.1080/14760584.2022.2100766.
Thank you for your comment, a metanalysis to estimate the pooled prevalence of vaccine acceptance has been added to the manuscript. (Figure 2)
Q4. In conclusions (Comparison with existing literature) you speak about "association", but your review cannot estimate an association. Please revise
The discussion section has been revised to exclude this statement. This is a useful comment. Thank you.
Q5. In introduction, you should describe the characteristics of the vaccination in adolescence (efficacy, effectiveness, immunogenicity, safety).
The introduction has been updated to incorporate this suggestion. (Line 40-44 and 51-55)

Reviewer 2 Report
I have read with interest this systematic review on the COVID-19 vaccination of adolescents.
This work adds to the current body of evidence; however, there are some issues that require further clarification.
1. Background
More details are needed on the COVID-19 vaccination coverage of adolescents and its associated factors. The authors could add indicative data on these issues and try to reinforce the justification of their work. I would rephrase the sentence on herd immunity.
2. Methods
The inclusion criteria should include mention of population, study design, and outcome.
In addition, I was not able to see any investigation of publication bias.
3. Results.
The authors may present the results by continent.
4. Discussion.
Future research needs may be described in the discussion section of the manuscript.
Minor editing is needed
Author Response
Thank you for your helpful comments.
Comments and Suggestions for Authors
This work adds to the current body of evidence; however, there are some issues that require further clarification.
- Background: More details are needed on the COVID-19 vaccination coverage of adolescents and its associated factors. The authors could add indicative data on these issues and try to reinforce the justification of their work.
I would rephrase the sentence on herd immunity.
Thank you for your feedback. The necessary amendments have been made to incorporate these changes. (Line 51-55)
We have rephrased the sentence on herd immunity.
- Methods: The inclusion criteria should include mention of population, study design, and outcome.
The criteria have been revised to include population, study design, and outcome. (Line 90-93)
In addition, I was not able to see any investigation of publication bias.
Thank you for your comment, further analysis with a funnel plot and Egger’s test was performed to investigate for publication bias. Egger’s regression test (p=0.545) did not indicate evidence of publication bias. (Line 166-167)
- Results: The authors may present the results by continent.
We have updated the table & figures to group the results by region.
- Discussion: Future research needs may be described in the discussion section of the manuscript.
Thank you for your feedback. The discussion has been revised to include this suggestion. (Line 304-307)

Reviewer 3 Report
Thank you for the chance to review this article. Please see below some comments.
Why did you combine 18-19 years old with 10-17 years old participants in one analysis? 18-19 are adult and they can decide by themselves.
The counting in PRISMA table is completely wrong, please re-check.
I found it difficult to read and understand Table 3 & 4. Please re-design.
Why was the Discussion structured? And why strengths and limitations are in the beginning of the Discussion? You need to discuss your findings first.
Section “Implications for practice and Research” has no link to your findings and does not reflect your findings. Update it or delete it.
I think the study design is not optimal and the conclusion does not have solid data. The analysis is very preliminary and does not give the reader any solid data.
It is difficult to follow the structure of paragraphs, some of them are one sentence (211-214).
Author Response
Thank you for your helpful comments.
Reviewer 3
Comments and Suggestions for Authors
Thank you for the chance to review this article. Please see below some comments.
Why did you combine 18-19 years old with 10-17 years old participants in one analysis? 18-19 are adult and they can decide by themselves.
Thank you for your comment. We have used the WHO definition which considers adolescents to be between 10-19 years of age. As alluded by the reviewer, we recognise the limitation in combining these age groups together and have updated the discussion section to reflect on these two seemingly separate subgroups of the study population. (Line 286-292)
The counting in PRISMA table is completely wrong, please re-check.
A total of 1,763 studies were identified with 689 duplicated removed, leaving 1,074 articles. The titles and abstracts of 1,074 articles were screened resulting in the exclusion of 1,403 articles. The full text was reviewed for the remaining 31 articles with 18 articles subsequently excluded. A total of 13 articles were included in this systematic review.
We have re-checked the numbers in the box and confirmed the numbers to be accurate. The diagram has also been revised to better place the arrows to indicate the various steps taken to identify the final included articles.
We would greatly appreciate it if the reviewer could kindly explain the miscalculations indicated in the comment for our rectification.
I found it difficult to read and understand Table 3 & 4. Please re-design.
Thank you for your feedback, the tables have been re-designed to improve readability.
Why was the Discussion structured? And why strengths and limitations are in the beginning of the Discussion? You need to discuss your findings first.
The discussion was structured to improve readability. We have relocated the “strengths and limitations” segment in the “Discussion” section.
Section “Implications for practice and Research” has no link to your findings and does not reflect your findings. Update it or delete it.
Thank you for your comment. We have revised to connect the findings with the implications for practice and research.
I think the study design is not optimal and the conclusion does not have solid data. The analysis is very preliminary and does not give the reader any solid data.
We have revised the manuscript to provide more details on the design and conduct of the systematic review, clarify the data and sharpen the analysis.
Comments on the Quality of English Language
It is difficult to follow the structure of paragraphs, some of them are one sentence (211-214).
This paragraph has been removed and the manuscript revised to improve clarity.

Round 2
Reviewer 1 Report
ok
Reviewer 2 Report
All my comments were adequately addressed by the authors.
Acceptable
Reviewer 3 Report
Including the meta analysis add value to the paper.